# Analysis of Driver's Reaction Behavior Using a Persuasion-Based IT Artefact

**Javier Goikoetxea Gonzalez [1],\*, Diego Casado-Mansilla [1,2],\* and Diego López-de-Ipiña [1,2],\***

[1]   Facultad de Ingeniería, Universidad de Deusto, Avda. Universidades, 24, 48007 Bilbao, Spain

[2]   DeustoTech–Facultad Ingeniería, Universidad de Deusto, Avda. Universidades, 24, 48007 Bilbao, Spain

\*   Correspondence: javier.goikoetxea@opendeusto.es (J.G.G.); dcasado@deusto.es (D.C.-M.); dipina@deusto.es (D.L.-d.-I.)

**Abstract:** The use of interactive technology to change behavior, which is commonly known as persuasive technology, is currently gaining attention in information systems research. It has been assessed in many application domains and the field of private mobility is not an exception, notably with the advent of self-driven cars. However, the reviewed body of research shows that when it comes to linking persuasion-based systems and mobility, most of the approaches focus on engaging drivers to use the car in a safer way, leaving the cost-efficiency aspect of driving less explored. Therefore, this article focuses on the study of a persuasion-based IT (Information Technology) artefact devised to make drivers more aware of car expenses (e.g., maintenance control, engine failures, enhance driving, etc.). Specifically, it aims to identify persuasive design principles for a smart IT solution that is tailored for the enhancement of the cost-efficiency of private cars. To this purpose, the results of a survey, where respondents (N = 301) were asked to rank different principles of persuasion which might result in increased efficiency to save time and money within their car, are presented. This work aims to contribute a persuasion-based IT artefact to help and influence drivers, enhancing their management of costs related to car mobility in real-time. The implications of the proposed solution, according to the responses of the survey, are discussed in line with its implementation and adoption by car holders.

**Keywords:** persuasive technologies; survey; mobility; motivation; behavior; IT artefact

## 1. Introduction

The growing technological development of the service sector and the globalization of markets, as well as the economy, are profoundly changing organizations' structures. Such a change is evident by the progressive incorporation into companies of the current emerging disruptive technologies (e.g., advances in robotics, information technology, Internet of Things (IoT), telecommunications, and artificial intelligence) [1]. These are very helpful when it comes to integration, innovation, and autonomy of the process towards digitalization. From an engineering perspective, Information Technology (IT) artefacts are usually designed to solve problems and, to a greater extent, produce a change in what is currently observable in order to generate a transformation in the behavior of users. Indeed, IT artefacts are not just the object of passive observation, but a potential context of transformation or intervention [2].

The car is still one of the most used means of transportation for short distances [3]. According to a recent industry study of Price Whaterhouse Coopers (PWC) [4], it is expected that current vehicles will have longer use by the year 2030, taking into account that 20% less vehicles will be sold. As a result, vehicles will have to withstand higher mileage contextualized on lower demand for new vehicles. If vehicles are going to have more use, the opportunity of gathering and analyzing vehicle information remotely, to manage cars' service needs, appears as a business opportunity. Indeed, this path opens

up a new research opportunity in terms of managing the car expenditure and, as a very important side effect, its generated pollution. We propose instrumenting cars with an IT artefact through which drivers are more conscious of the actual impact of their driving behavior and which is recommended for better practices to save fuel or avoid unnecessary consumption of fuel. Therefore, the end result is that the car sustainability, in terms of economic and environmental aspects, can be improved.

Specifically, one of the elements that needs to be analyzed further is how the persuasion-based strategies through Information Communication Technology (ICT), which have been applied with success in other contexts [5–7], affect car expenses. Indeed, Paraschivoiu et al. recently reported that persuasive applications targeting driving attitude and other aspects, apart from safety and eco-driving, are almost entirely lacking [8]. The basis of this research is that cars are driven by people and those people could be persuaded in different manners [9]. Following this perspective, in this work, the influence of persuasive technology on drivers is analyzed. In addition, it aims to understand what persuasive principles are most relevant when they come to reduce the car mobility costs.

Continuous developments are being made to improve the efficiency of travelling, by reducing time consumption, alongside making it safer for the general public [10]. In parallel, various systems, tools and devices are designated to enhance not only mass communications but interpersonal communications [11]. The development of smartphones has greatly helped users in terms of navigation, thereby making travelling much more reliable nowadays [12].

With these premises, there is no doubt that technology, through interactive systems, may play different roles in relation to human beings [13]. The interaction between the two can be presented in three ways: assigning the computer the role of tool, media, and social actor. The role of technology, as a social actor, is one of the most important persuasive approaches in the Human–Computer Interaction (HCI) model [14]. Through this role, it is possible to influence people through positive feedback, helping to improve behavior, and providing social support. In this context, "persuasive technology" refers to the study of the ability of technology to convince. Specifically, persuasive technology aims at changing, forming, or removing behaviors by interacting with ICT-based systems. Fogg [9] coined the concept of "captology", i.e., the study of how computers can be used to persuade people to change their attitudes or behaviors. Alternatively, Oinas-Kukkonen and Harjumaa [15] (p. 486), defined a persuasive system as a "computerized software or information system designed to change attitudes or behaviour (or both) without using coercion or deception". Thus, building upon psychological research on human persuasion, it seems that a set of persuasive design principles can be embedded into IT artefacts to persuade users to engage in a target behavior [16].

Based on the above reasoning, this research work analyzes an existing IT artefact for the car. The artefact is an IT system connected to the car that provides relevant information that can be used in different ways by the drivers (e.g., maintenance control, engine failures, enhance driving, etc.). As such, the IT artefact takes into account that the reaction of a driver in each context and moment can be influenced by many factors from the design of the IT artefact itself to the state in which the driver is in [16]. In this context, Fogg [16] provided three determining elements to form, maintain or spark a behavior: (a) the motivation of the individual; (b) the ability or ease of the actions proposed and (c) the existence of triggers to bring the behavior to occur. Hence, the design of the IT artefact draws upon this triad to form or enhance a driver's eco-minded behavior.

There have been different approaches concerning the introduction of an IT artefact connected to the car. Many of them are concentrated on explaining issues related to safety and security [17] or maintenance and remote management of the car [18]. We have identified an existing gap in explaining monetization and savings around the car usage with a sustainable analysis focus. In this sense, reduction in money expenditure around the car could be translated into having a more sustainable behavior around the car usage. Related to this, the IT artefact recommends actions, services and products oriented to improve the driver decisions with the final objective of reducing car expenses and in some cases the car usage. These recommendations could reduce the number of cars driving on the

roads with a clear sustainability impact in terms of environment effect, improving the $CO_2$ emissions as well.

Taking the IT artefact as the basis, the goals of the article are threefold: (i) to analyze the impact of the different driving services and how they can be helpful to the driver decision-making process (according to Duncan et al. [19] people that have a driver assistance system usually provide positive opinions about other systems such as navigation); (ii) to examine the influence of the persuasive technology on drivers and driving performance; (iii) to underpin which persuasive principles, among those provided in the existing literature, are most applicable to drivers when it comes to reducing the economic costs around the car.

The work is divided into four parts. Firstly, in Section 2, the context of the research and the used artefact are described. Secondly, the related work is reviewed. The origin, evolution, and importance of advances and innovations are discussed to review relevant bibliography on the areas of car services and Persuasive Systems Design. Thirdly, the methodology used in this article, to try to identify if technology plays a relevant role with drivers and which principles of persuasion are more relevant than others in influencing a driver, is detailed. Fourthly, the results of a questionnaire applied to a wider range of drivers are analyzed in Section 5. Finally, Section 6 concludes the article with some conclusions, recommendations, and future work.

## 2. Context of the Research: The Driver Artefact System (DAS)

As a departure point, an IT artefact available in the market is used. This artefact or Driver Artefact System (DAS) is provided by Grupo NEXT (www.gruponext.es), a Spanish-based company. This company is focused on mobility data management to foster cost-efficiency driving models. The IT artefact selected has three main elements. The first element is (i) an OBD-II device (On Board Diagnostic)—a standard device connected to the vehicle, provided with a Subscribers Identify Module (SIM) card inside that transmits mobility information (e.g., location, speed, acceleration, braking, etc.) as well as information from the car's on-board computer (e.g., engine failures, engine temperature, maintenance needs, etc.). The second element is (ii) the cloud-based platform, where mobility data are collected, and the information is analyzed. Finally, the last element of the DAS is (iii) the end user mobile Application (APP). With this APP, the driver can browse his/her ongoing activity. Figure 1 illustrates the different components framing the IT artefact.

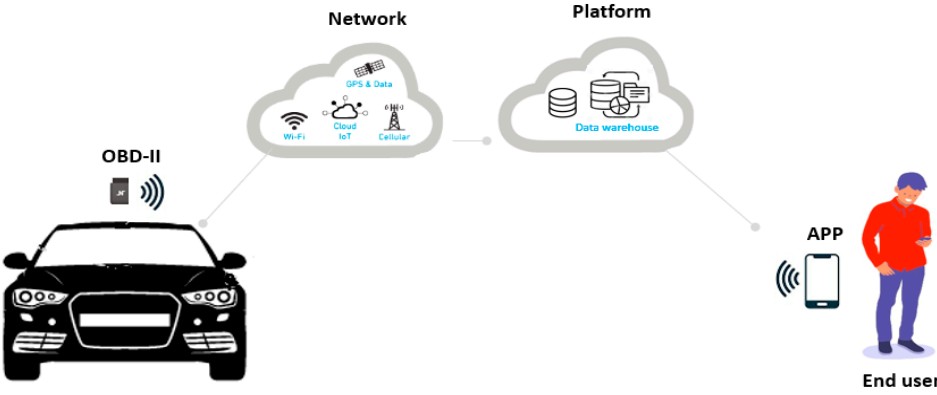

**Figure 1.** High-level diagram of the overall system.

The architecture of the DAS provided, including details of the internal components of its three layers, is depicted in Figure 2. Those components are:

(1) The Platform is a high-performance computer system, equipped with redundancy characteristics in all its critical elements, to allow the provision of the service without temporary interruptions.

The platform also has interfaces with the information systems of the partners that provide real time services to drivers.

(2) The On Board Diagnostic (OBD) device, connected to the car port of each of the vehicles, reads data from the various sensors that the car has to monitor the status of its various components and to send them to the platform for processing and exposure (ordered and managed). Generically, we can call these data the telemetry of the vehicle [20]. In addition, the device also incorporates a component to determine the geographical position in which the vehicle is using GPS technology. The most common electronic components of this type of device are a GPS module, a Global System for Mobile (GSM) communications system, an accelerometer, a gyroscope, a small microprocessor, the diagnostic process and a backup battery [21]. The data are collected from the OBD device that is part of the IT artefact, which will allow users to know what is happening in the car and its mobility in real time and, thus, approach the user to propose a maintenance solution, whenever needed. The DAS is a self-installing device, which means that no professional installers are needed to deploy the device in the vehicle. This simplifies the testing. All these data are sent to a central computer to be analyzed and processed thanks to the SIM installed inside the artefact. The device contains a SIM card which, conveniently activated in the operator's 3G/4G network, facilitates the periodic sending of car info. The device is complemented by the creation of a communications protocol. This communications protocol is used to send DAS info to a server.

(3) A mobile application allows the driver to manage the system/service himself, from its activation to its deactivation—including its configuration, the subscription of new value-added services and some data visualization. It also offers information about the saved money in real time, for car management purposes. This mobile application must be available for both Android terminals (versions equal to or greater than 4.4.x-KitKat) and iOS (versions equal to or greater than 9.0). The granularity of data collection as well as the frequency of delivery to the server can be handled manually by the operator of the artefact. According to Fogg [9], the APP, as end-user endpoint system, is a critical factor for the success of DAS.

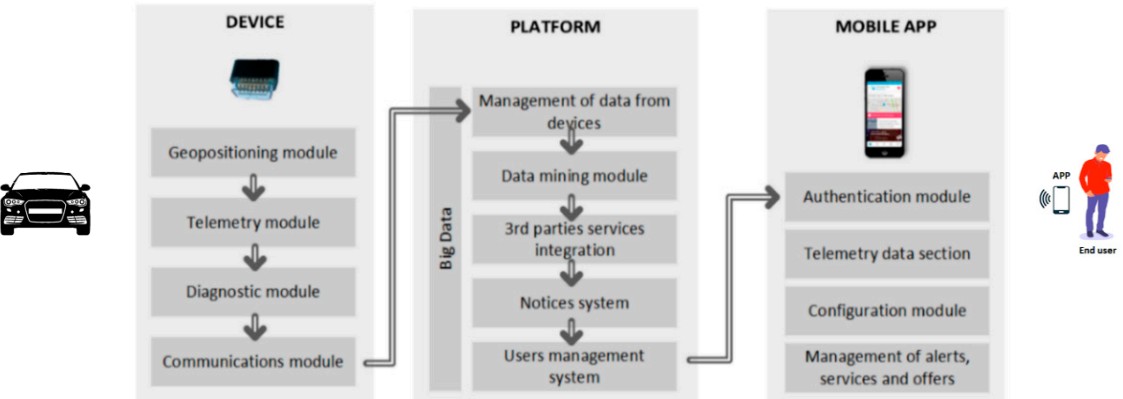

**Figure 2.** Technical IT artefact solution high-level diagram.

The APP information is organized in two categories: (i) car information (location, speed, car failures, engine check status, car movements, fuel, trip statistics, maintenance schedule, insurance schedule) and (ii) driver services through customized information and suggested promotions in real time (parking, restaurants, gyms, hotels, and so on). The first category refers to car events, naturally recorded during the vehicle mobility. The second category refers to real-time driver needs detection and offers in order to provide drivers with the best answers in terms of products and services. This second category is the target of this work, since the messages offered by the DAS in real-time might be much more efficient by using more appropriate persuasion strategies.

Figure 3 depicts some APP screenshots. The first one refers to the capacity of the system to send customized push messages to the end user's phone. The second screenshot offers a customized proposal (once the end user has shown interest in the push message by touching over it). The third image shows the alarms configuration, where the end user can activate or deactivate the information that she wants to receive. The fourth screenshot refers to car and driver personalized information. In the fifth one, the APP home screen can be seen, which includes some basic information and the savings achieved up to now. Finally, the last screenshot shows information related to trips: trip consumptions and trip details.

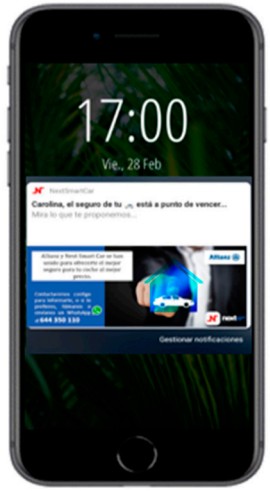

1. Customized push messages

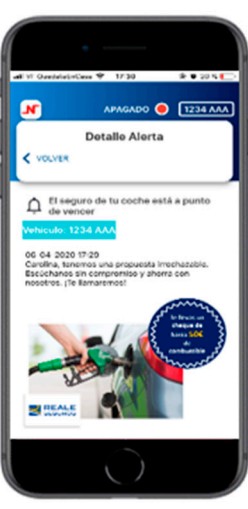

2. Suggested promotions

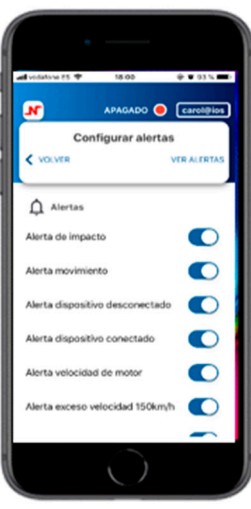

3. Alarm configurations

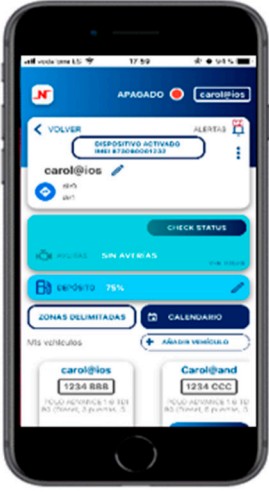

4. Customized driver information

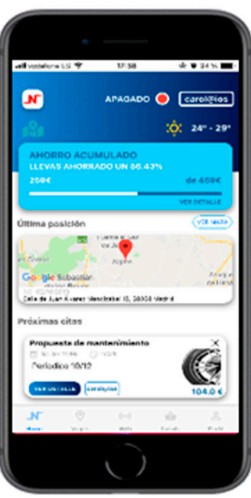

5. Location and customized message

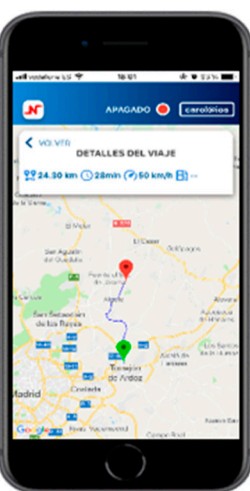

6. Trips and road information

**Figure 3.** Most relevant Application (APP) screen shots.

The three previously mentioned components, and the screenshots, considered technical aspects of efficient design. As with any element of automotive use, they must go through rigorous evaluation and testing methods to meet the required specifications [22].

Currently, cars are equipped with many electronic sensors that fulfil several functions from receiving and issuing a signal to automation through the permanent exchange of data and information. Therefore, this explains the importance of this study and literature review of the existing research.

## 3. Background and Literature Review

This section reviews the existing literature on the design of effective systems based on persuasion to understand how an IT artefact can be devised to nudge drivers towards a more cost-sustainable driving. The context of sustainability is based on cost-efficiency and real-time services, depending on the car or driver needs. The impact of various driving assistance systems on humans is also reviewed and how they can be helpful in the driver decision-making process.

The revision of the body of knowledge starts with the Persuasion Design Principles. Figure 4 shows the literature review process. As can be observed, four categories of analysis have been performed: (i) Persuasive Design Principles to understand the framework; (ii) Persuasive Design Principles that apply to this research; (iii) technology that could persuade or influence a driver; (iv) the acceptance level of drivers with regards to different technologies. The idea was to study the different existing technologies to persuade or influence people and more concretely, if it was possible, understand the driver's reaction. In essence, this section overviews the theoretical basis of the study and reviews the impact that various driving assistance systems has on humans and how they can be helpful in the driver decision-making process when it comes to reduced the car expenditure.

**HCI**: Computer Human Interaction; **ADAS:** Advanced Driver Assistance System; **ITS:** Intelligent Transport System; **MIS:** Management Information System; **TAM:** Technological Acceptance Model; **TRA:** Theory of Reasoned Action; **TPB:** Theory of Planned Behaviour; **TRAM:** Technological Readiness Acceptance Model; **DAS:** Driver Artefact System

**Figure 4.** Literature review workflow.

According to De Steno et al. [23], persuasion attempts are more successful when the behavior released by the message is in harmony with the emotional state of the recipient. The use of different emotions has different levels of influence [24]. The primary purpose of being able to evaluate the driver's interaction, with our IT artefact, is to observe the behavior of the drivers, to produce appropriate responses in different situations [24].

Coming back to the theory behind this study, it is necessary to address the persuasion-based investigations. Research in psychology has long studied persuasion to foster behavioral change [19–21]. People's behaviors have been studied in terms of the influence that the use of technology can have on them [10,22]. However, they have not studied the impact, in terms of savings, that a person can have based on the influence or persuasion of a machine in real time. According to Oinas-Kukkonen and Harjumaa, [15] there are four types of persuasive categories, with 28 principles. In addition to these

four persuasive categories, Fogg [16] (p. 187) introduced "the Kairos Factor". This factor states that: Thanks to a mobile device (the IT artefact), we can present a message at the right moment to increase the potential to persuade [16] (p. 188).

Besides persuasion, the field of human–machine interaction has also been reviewed to understand the appropriateness of the technology to the driver and the interaction design that should be created. Duncan [25] discussed that if a person has a driving-aid system, she will become more familiar with the technology that is behind it and therefore this person will have a better opinion of it. In other words, people adapt to technology, although ideally technology should adapt to people. In Table 1, a summary of the literary review of different technology elements that can influence driver persuasion is presented. On the leftmost column of the table, an acronym corresponding to the different systems analyzed is provided. The second column is related to the description, that is, the description is the full acronym name. The Notification System column indicates the way that technology acts to try to influence the driver (the interaction modality). The fourth column makes reference to the purpose of each persuasive approach; thus, the goal (e.g., influence on human behavior, driving safer, driving more efficiently, improve driving, real-time recommendations, etc.). The fifth column captures the technology system in which the driver's behavior is impacted. Finally, on the rightmost column, the bibliography reference for each approach analyzed is included. This schema resembles the one proposed by Paraschivoiu et al. [8] who presented a review of persuasive systems in vehicles based on the Persuasion Interface Design in the Automotive context. Their framework integrates intents (the goals), cues (the way to persuade), persuasive principles and design options (notification system and technology used) for automotive persuasion.

**Table 1.** Different approaches to persuading a driver in the field of mobility.

| Acronym | Description | Notification System | Designed to | Technology | Article Reference |
|---|---|---|---|---|---|
| PDP | Persuasive Design Principles | Theory | Influence human behavior | General | [16] |
| HCI | Computer Human Interaction | Multimodal notification interface | Driving safer, comfortable and efficient | Embedded into the car | [26] |
| ADAS | Advanced Driver Assistance System | Multimodal notification interface | Safety and Efficiency Driving | Embedded into the car | [27] |
| ITS | Intelligent Transportation Systems | Road panel notifications | Improve drive safety | Road panels | [28] |
| MIS | Management Information System | Road panel notifications | Information (places, petrol stations, … ) | Road panels | [29] |
| TAM | Technological Acceptance Model | Theory | Technology use acceptance | General | [30,31] |
| TAM2 | Technological Acceptance Model 2 | Theory | Technology use acceptance | General | [32] |
| TAM3 | Technological Acceptance Model 3 | Theory | Technology use acceptance | General | [33] |
| TRA | Theory of Reasoned Action | Theory–TAM predecessor | Technology use acceptance | General | [34] |
| TRAM | Technological Readiness Acceptance Model | TAM evolution | Technology use acceptance | General | [35,36] |
| DAS | Driver Artefact System | Based on OBD-II device | Real-time recommendations | OBD and APP | [37] |

In Table 1, eleven types of technological ingredients that can be used to analyze the influence of an IT artefact on people and, more particularly, to influence drivers, were analyzed. The application of the Principles of Persuasion in drivers was examined. Furthermore, different systems and technologies related to interacting with drivers, inside and outside the car, are addressed. In this context, Human–Computer Interaction (HCI), as a multimodal notification interface designed to facilitate driving [26], has been reviewed. The Advanced Driver Assistance System (ADAS) was also included

because it is a technology system for safety notification interfaces in real time while driving [27]. In order to consider ever more prevalent road infrastructure notification panels, Intelligent Transport Systems (ITSs) were also analyzed. They correspond to a set of road infrastructure solutions that are designed to improve the safety and comfort of drivers and passengers [28]. The literature referring to demand/response to incentives with the aim to achieve economic benefits was also included in the study of the existing research. In this field, the Management Information System (MIS) is the technology model analyzed. This technology is oriented to causing changes in a driver's behavior with respect to a usual pattern of consumption, in response to price signals or incentives on roads [29].

The acceptance of new technologies through TAM (Technological Acceptance Model) was also considered. TAM is a theory of information systems that models how users accept and use a technology [30,31]. TAM is analyzed, as well as its predecessor, namely, Theory of Reasoned Action (TRA) [34]. Attempts have been made to develop several derivations of this model, namely, TAM 2 [32]; TAM 3 [33]; Technology Readiness and Acceptance Model (TRAM) [35,36]. Chaumon [38] describes the process of acceptance of technology as a trajectory that goes from the beginning of the design of a system until its implementation. From this notion, different authors [34,35] have identified two important stages: The preadoption stage, which corresponds to social acceptability and the subsequent stage, which corresponds to practical acceptance. Finally the acceptance associated with the use of evaluating an IT artefact in real-time has been studied as well [37].

At the bottom of the table, the DAS (Driver Artefact System) system analyzed by this paper's authors has been included. As explained in Section 2, DAS is a system that, depending on the drivers' taxonomy and driving patterns, based on log analysis, offers an ad-hoc solution in real-time through a technological artefact.

Up to now, in this review, it is observed that there are already many technological elements designed to influence drivers' behavior. Most of them were devised to make technology more accessible, more understandable, and even simpler to drivers. Some of them, in terms of mobility, try to help the driver to be more efficient and safer on the road. However, to the best of the authors' knowledge, there is no existing research that employs ICT-based artefacts to help and influence drivers to save time and money in his/her car mobility activities in real-time. More concretely, no evidence has been found across the bibliographical review that refers to what kind of persuasion principles would apply better to each driver in different contexts.

## 4. Methodology/Procedure

This section explains the methodology framework as well as how it fulfils the goals that were stated in the introduction section. The methodology approach consists of using the DAS artefact to collect real-time data and to use such data to influence drivers towards saving money around car ad hoc needs (e.g., maintenance control, engine failures, enhance driving, etc.). As explained in previous sections, recall that DAS allows users to know what is happening in the car and assists the driver to propose a maintenance/service solution is required. To evaluate driver's behavior, the study of persuasive technologies is applied to understand which technological advantages can be used to motivate end-users. To such extent, a survey was created. Through this instrument, the respondents had to determine which of the persuasion principles presented were more relevant for them. Instead of asking users about the twenty-eight persuasive principles identified in the literature, only fifteen were included in this study. This decision was taken based on the reasoning of experts on persuasion and followed the same agreement-based rationale taken in a previous paper of the authors [5].

The survey was launched in an online format. People who received and then answered the questionnaire were selected because they already had the DAS installed in their cars. Those users had been using DAS for 7 months before receiving the survey. Hence, those drivers were already familiarized with the system and knew the functionalities and services provided. Besides, 55% of drivers (end users) do use the APP on a daily basis, 65% weekly and more than 80% monthly. Drivers open 85%, on average, of the customized push messages and go to the APP promotion zone to get

more information. Therefore, it can be stated that a wide percentage of DAS adopters are active users. Apart from this, the questions in the survey were ordered randomly to remove any possible bias or mechanic introduction of answers, i.e., requiring the users to pay a higher attention to each of the questions posed. The access to those drivers has been possible thanks to an agreement with the company Grupo NEXT. The survey was sent to 1500 drivers randomly in a blind selection process and after two weeks 308 responses were obtained. Of these, 301 of the responses were valid after a cleaning process and outliers removed. The data protection guidelines, the legal GDPR and anonymization policies, were first consented to by participants before starting the questionnaire. Following this, the survey was launched once the University of Deusto's Ethics Commission had approved the plan. Everything was performed in accordance with the university's rules and as mentioned, following the Data Protection law. It is important to emphasize that the authors of this paper did not have access to neither private information nor to the identity of the drivers. The respondents were informed about the purpose of the questionnaire and about its scientific use. Finally, users were informed that their participation was voluntary. Some reminders were sent during the survey period to try to ensure an adequate participation rate.

In the survey, the end-users were asked to self-rate (from 1 to 7) strategies derived from the principles of persuasion that apply to this research. For each persuasion principle, a strategy was designed (see Table 2 where a mapping between the Persuasion Principles and the strategies of persuasion is outlined).

**Table 2.** Mapping of persuasive principles and strategies.

| Persuasion Principle | Description | Strategy | Organize |
|---|---|---|---|
| P1. Principle of Tunneling | It provides opportunities to persuade the user along the way. In our case, we use our IT artefact to persuade the end user to do things at every single opportunity. | The APP is available everywhere at anytime | S1 |
| P2. Principle of Tailoring | The information provided by the IT artefact is more precise if it is bespoke to end users. In our case, the communication with the end user will be optimized depending on the categorization. | The APP offers me "tailor-made" services | S2 |
| P3. Principle of Personalization | The system is modified to meet the user's needs. A system that offers personalized content or services has a greater capability for persuasion. In our case, drivers will receive push notification messages (QR codes) through the IT artefact that can be presented on the suggested places (shops) to have rebates and discounts. | The APP can be customized by the user | S3 |
| P4. Principle of Self-Monitoring | The goal is to allow people to monitor themselves in order to modify their behaviors, to achieve a predetermined objective or outcome. In our case, drivers will receive full information of their savings through an APP to check savings account. | The APP allows end users to monitor the behavior and allows them to better reach the goals (control, savings, etc.) | S4 |
| P5. Principle of Simulation | Related with cause and effect. In our case, depending on the end user circumstances and classification taxonomy, opportunities will happen. The more information we have, the better opportunities we can suggest. | The more I use the APP, the more proposals I receive | S5 |
| P6. Principle of Praise | Depending on the praise, DAS system can make users more open to persuasion | Users receive praise through the APP | S6 |
| P7. Principle of Rewards | The system could reward depending on the behavior. Using our IT artefact, we suggest different reward systems focused on discounts and rebates. | The APP allows me to obtain savings | S7 |
| P8. Principle of Reminders | The more you remind someone of something, the more the users will achieve their goals. To persuade the user along the way, we send some reminders to try to influence the end user. | End users appreciate that the APP reminds them of things that they can do or things that they can have | S8 |
| P9. Principle of Suggestion | In an interactive computer product, it corresponds to its ability to suggest behavior at an opportune moment. Application for a driver with real-time solutions to car services with savings. | Proposals and suggestions come to me at the right time | S9 |
| P10. Principle of Liking | Depending on the IT artefact's visual attractiveness, we can be more persuasive. We are using digital marketing tools to persuade the end user. | The way in which the proposals come to me condition my response | S10 |

**Table 2.** *Cont.*

| Persuasion Principle | Description | Strategy | Organize |
|---|---|---|---|
| P11. Principle of Expertise | A system that is viewed as incorporating some level of expertise will have more persuasive power. In our case, we will use some artificial intelligence to describe the end user ontology to try to be more precise in suggesting product and services. | End users watch the APP as a sophisticated system and that encourages them to use it more | S11 |
| P12. Principle of Authority | System should refer to end user with some level of authority. In our scenario, we can suggest driving scores to try to influence on their driving risk. | The APP suggests to me comments or actions that I take very seriously | S12 |
| P13. Principle of Social Learning | The end user would be more motivated to perform our target behavior if the system will provide info about others performing the behavior. In our case, thanks to the gamification options, the end user will be compared over their peers in some mobility and driving aspects. | The APP allows me to compare myself with other users | S13 |
| P14. Principle of Recognition | Offering private, public, individual or group recognition can increase the probability of targeting a behavior. In our case, the targeting behavior is a critical part of the IT artefact solution. | The APP shows me a recognition or scoring system | S14 |
| P15. Principle of Kairos | The mobile systems of the future will be able to identify opportune moments and have more effective influences than they do today. The artefact will show drivers products and services, depending on the needs and circumstances at the right moment. | The APP helps me to identify opportunities and services at the right moment | S15 |

The selection of the strategies on top of the fifteen persuasion principles was influenced by scholars' views in the field [9,10,34] and the authors' own criteria. As can be observed, purposely all the strategies are related to the use of the APP of the DAS system.

After testing the questionnaire with some pre-test participants, the authors decided that some of the strategies and principles could be grouped to facilitate the selection and, hence, ensure comprehensibility of the whole approach. For the same reason, the survey questions were simplified. Due to the drivers' prior knowledge of the system, questions were formulated in a basic way, since users already had the context about the APP and the car artefact. The main idea of the study was to understand how drivers rank the different persuasive strategies, more than discovering new functionalities. The Hawthorne effect, in which individuals modify an aspect of their behavior in response to their awareness of being observed, was removed during the first and second month over the use of the DAS. Therefore, the responses received were freed of such conditioning effect.

Therefore, some principles and strategies were grouped together. Table 3 shows the strategies clustered into 7 "main strategies".

**Table 3.** Mapping of Persuasive Strategies to the Persuasion Principles.

| Strategy | Definition/Question | Persuasion Principle | Question |
|---|---|---|---|
| S1 | The APP is suitable anytime in the day | P1 | 1 |
| S2, S3, S15 | The APP offers me "tailor-made" services and can be customized by the user | P2, P3, P15 | 2 |
| S4 | The APP allows end users to monitor the behaviour and allows end users to better reach the goals (control, savings, …) | P4 | 3 |
| S5, S6, S7 | The more I use the APP, the more proposals I receive where I like to receive praises for getting savings | P5, P6, P7 | 4 |
| S8, S9 | I appreciate that the APP reminds me things I can do or things that I have to do in real time | P8, P9 | 5 |
| S10, S11, S12 | The way in which the proposals, as a sophisticated system, come to me, conditions my response | P10, P11, P12 | 6 |
| S13, S14 | The APP shows me a recognition and allows me to compare with other users | P13, P14 | 7 |

Table 3 explains the relationship between the questionnaire and the definitions/Persuasive Principles/questions. This means that question 1 refers to strategy 1 (S1), question 2 refers to strategies S2, S3 and S15, question 3 refers to strategy S4, and so forth.

The questions used in the survey are included in Table 4. To aid the table understanding, two columns have been included: One with a short statement reused in some graphics in the results section and a second column with the screenshot equivalence presented in Figure 3, fourth column. Respondents were asked, firstly, for a prioritization task in order to identify which of the strategies were most important to them and which were less relevant according to their criteria. Specifically, responders were asked to identify and rate the functionalities that the APP (part of the DAS system) offers with the aim of organizing them from the lowest to the highest relevance. The criterium was to give a score of "1" to the strategy that the users least identify with, or s/he liked the least. On the opposite, a score of "7" was requested for the strategy/functionality that one thinks that can help him/her to use the app more and/or save more time and money. The objective was not to repeat any score in the seven answers in a way the questions could be ranked. The questions were shuffled to avoid bias in the responses.

**Table 4.** Survey questions, Short Statement, and equivalence of APP Screen Shot.

| N° | Question | Short Statement | Screen Shot |
|----|----------|-----------------|-------------|
| 1 | The APP is suitable for all times of the day | Time suitable | 4 |
| 2 | The APP offers me "tailor-made" services and can be customized by the user | Tailor-made services | 2, 3 |
| 3 | The APP allows the end user to monitor the behaviour and to better reach the goals | Reach goals | 4, 1 |
| 4 | The more I use the APP, the more proposals I receive where I like to receive praises for getting savings | Receive praises | 5 |
| 5 | I appreciate that the APP reminds me things I can do or things that I have to do in real time | Remind things | 1 |
| 6 | The way in which the proposals, as a sophisticated system, come to me, conditions my response | Response conditioned | 4, 6 |
| 7 | The APP shows me a recognition and allows me to compare with other users | Users comparison | 5 |

## 5. Results

To find a visual and simple way to interpret in a glimpse which strategies were the most interesting for the respondents, answers were weighted for each question. The weight system was created following this scheme: If a respondent rated the first question with the highest rank, such answer was multiplied by a factor of seven. If a respondent rated the second question with the second highest rank, that answer was multiplied by a factor of six and so on until reaching the last question that was multiplied by a factor of one, being the strategy of least interest to the respondent. The same questions are always multiplied by the same factors. With this weighting system, and by aggregating the rankings for each question from the 301 respondents, the results of the survey can be provided graphically.

As can be observed in Figure 5, answers 3, 2 and 5 are the ones that have obtained the best rankings. Strategy number 3 obtained 17.69% of the answers, number 2–17.47% and number 5–15.39%. Those strategies (3, 2 and 5) refer analogously to: offering personalized services, allowing the car behavior monitoring to reach the objective of saving money, and providing solutions and services in real time. In fact, the dominant strategies with higher rankings among respondents were those referring to personalized consumption control. The least important or lowest ranked were those related with public recognition (question 7) and the way in which the answers arrived to the APP (question 6).

The type of functionality provided influences the respondents' assessments but most of them prefer strategies 3, 2 and 5. It is very clear that those strategies are better valued than the rest. As previously mentioned, the questions in the survey were ordered randomly to remove any possible bias.

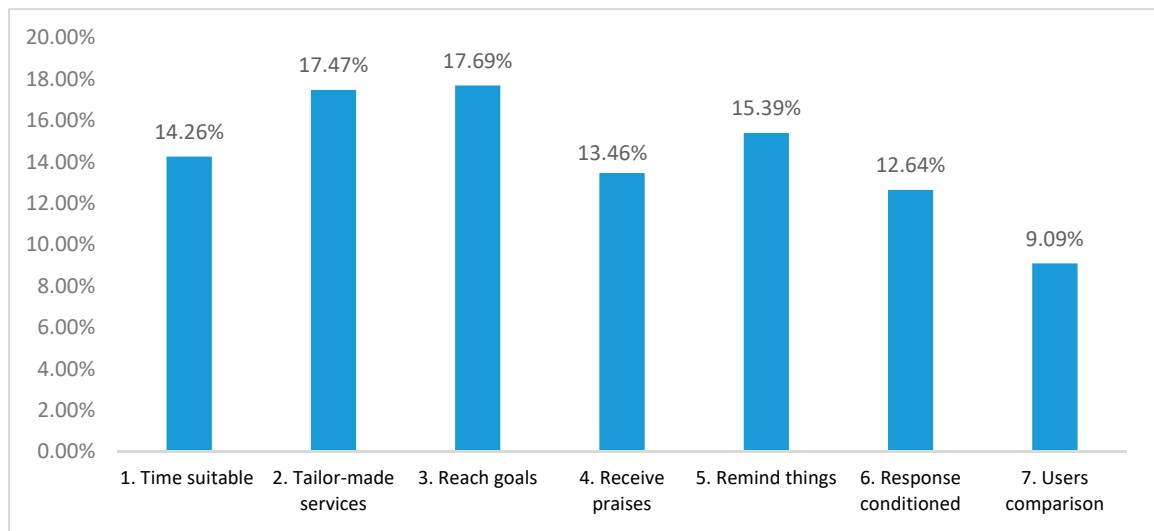

**Figure 5.** Weighted answers from the questionnaires.

To see if the results vary applying a different weighting factor to the rankings, Figure 6 is provided. It shows the results with only the best and worst ranking to different questions taken into account. Thus, all other rankings are overlooked focusing only on the strategies the respondents prefer the most and the least. In this second approach, the strategies with the highest level of acceptance are number 3 and number 2 (personalized services and self-monitoring). Those with the lowest level of acceptance are number 7 and number 6. These last two answers are referred to: (i) if the end user is conditioned depending on the way in which she receives offers, and (ii) if the DAS shows to the driver a recognition system to allows his/her to compare with others.

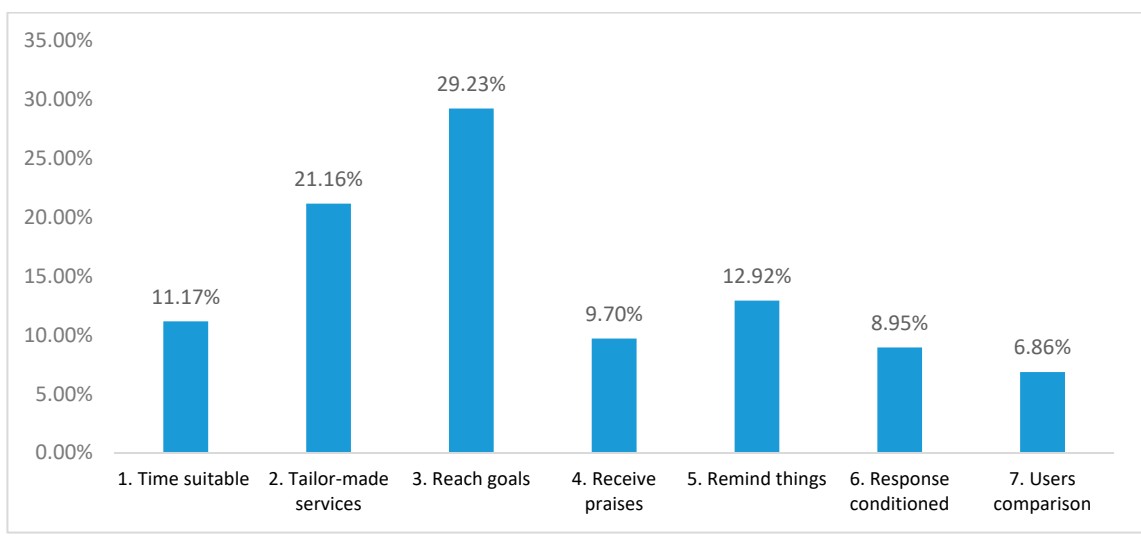

**Figure 6.** Best (7) and worst (1) answers ranked for comparative purposes.

In terms of Persuasion Principles, as can be observed in the Table 5, the top six principles that were most highly rated are related with the Principle of Tailoring, the Principle of Personalization, the Principe of Kairos, the Principle of Self-Monitoring, the Principle of Reminders and the Principle of Suggestion. Overall, this analysis indicates that the most important value of the different Persuasion Principles is related to personalized recommendations to save time and money through the APP.

**Table 5.** Best evaluated strategies.

| Ranking | Question | Strategy | Principle | Principle Description |
|---------|----------|----------|-----------|---------------------|
| 1 | Question 3 | S4 | 4 | Self-Monitoring |
| 2 | Question 2 | S2, S3, S15 | 2, 3, 15 | Tailoring, Personalization, Kairos |
| 3 | Question 5 | S8, S9 | 8, 9 | Reminders, Suggestion |

The messages sent by DAS are fully customizable and the key is to deliver those messages through the best persuasion strategies. These messages sent at the right time, through the right communication channel, can condition users' responses. Those responses could be translated into purchase stimulus. By conditioning the stimuli, it can therefore alter drivers' decisions in terms of consumption.

By observing the impact of the technology on drivers, it can be concluded that Persuasion Principles are effective to engaging driver behaviors, mainly focused on preparing customized messages, on time, in form and with a large cause–effect relationship. It could also be argued that it is important to know that people are ready to admit recommendations if those recommendations are useful to make life easier for drivers.

One of the objectives of this article was to discover which persuasion principles are more relevant towards designing a better IT artefact. Progressing towards a better artefact implies increasing its use by offering increased and better real-time solutions, which impact the drivers by enabling them to save money around the car mobility expenses. The best evaluated strategies are indeed a clear pointer to develop a better IT artefact.

The results in terms of the worst evaluated Principles of Persuasions can be observed in Table 6. The worst ranked were the Principle of Social Learning, Principle of Recognition, Principle of Linking, Principle of Expertise, Principle of Authority, Principle of Simulation, Principle of Praise and finally, the Principle of Rewards. It is necessary to point out that this article has a limitation because respondents were asked to express his/her opinion over an already existing IT artefact. Such artefact does not still feature functionalities related to the capacity of providing rewards, recognition and/or liking. Therefore, the worst evaluated strategies seem to be those which are not yet available in the current version of DAS. This is a relevant issue since an otherwise deployed IT artefact feature could alter the presented results.

**Table 6.** Worst evaluated strategies.

| Ranking | Question | Strategy | Principle | Principle Description |
|---------|----------|----------|-----------|---------------------|
| 1 | Question 7 | S13, S14 | 13, 14 | Social Learning, Recognition |
| 2 | Question 6 | S10, S11, S12 | 10, 11, 12 | Liking, Expertise, Authority |
| 3 | Question 4 | S5, S6, S7 | 5, 6, 7 | Simulation, Praise, Rewards |

## 6. Conclusions, Recommendations, and Future Work

This paper has studied and analyzed the implementation of persuasive design principles in a real use case, i.e., NEXT company's DAS device. This work has assessed the impact that persuasive technologies may have on a user's behavior. As a result, a selection of the best persuasion strategies to aid drivers has been performed. According to the definition of persuasive technology which considers any interactive computer system designed to change the attitude or behavior of a person [16], the analyzed DAS devices can be considered an element of persuasive technologies.

Better strategies could be translated into creating better IT artefacts to produce better results in terms of solutions to real needs. The way to present those products or services is critical. Responders have helped to identify the right strategies to design a better DAS artefact. In that context, there is a line of research for future work to create better customized IT solutions based on those Persuasion Principles.

This article oriented to select the best persuasion strategies could be the basis for an effective way of reducing costs or enhancing sustainability in different areas. In this sense, this work has found

how to better design an IT artefact for the car industry to impact the end user and invite him/her to be more sustainable in terms of saving money and time. According to this study's results, a key recommendation for creating a new APP or improving the current one, would be to augment the personalized messages to alert the driver. Other recommendations oriented to produce a better answer whilst driving supported by an IT artefact could be to improve the system to better monitor drivers themselves, to be very precise with the information generated by the system, to be able to send messages in real time and to remind users to do things to achieve their goals. All of those strategies are related to the Principle of Tailoring, the Principle of Personalization, the Principle of Kairos, the Principle of Self-Monitoring, the Principle of Reminder and the Principle of Suggestion. DAS will be enhanced by implementing mechanisms to promote the most effective Persuasion Principles Strategies that have been identified in the questionnaire.

On the other hand, it is also important to observe that the participants in the survey were people with a high confidence in technology because they were using the device on a daily-based. Future steps will seek to understand the differences between people that currently are using technology in their cars in comparison with people that do not. This could condition user responses. Along with this possible weakness, it is often discussed whether technology can condition people or if it is just a tool that makes life easier for users. We deem that whoever creates the technology has the power to add elements that could condition users and, therefore, there is an ethical side to the development of the technology itself. However, persuasive technology could be considered as a means to nudge end users without using coercion. Notably, technology has progressed from being a tool "without effect" to an element that can condition people's reactions. It is therefore paramount to address ethical and moral questions if acceptance of technology in our daily lives wants to be enhanced, as we pursue in this work. For example, is it ethical to design a device that can contribute to modify people's behavior? In our view, it is as long people's compliance has been previously ensured. Additionally, we acknowledge that this study has some limitations; i.e., we have little information about the socio-demographic profiles of survey responders, even it is certainly a representative sample, the truth is that it does not cover the entire population that uses the artefact.

With regards to future work, DAS can be also improved with novel persuasion strategies. This will be aligned with fewer resources used, money saved and overall reduction in $CO_2$ emissions. However, evaluating the impact of these strategies is considered as future work. Besides, our approach to select the best Persuasive Principles to create car IT artefacts for giving answers to real-time needs, could also be applied to other areas such as home supplies (electricity, heating, water, etc.) and personal expenses (hotels restaurants, trips, etc.) to mention only two options. Additionally, developing a driver's taxonomy could also open a relevant work horizon since this research is oriented towards more personalized technological elements based on AI that allows a higher degree of personalized solutions.

**Author Contributions:** Conceptualization, J.G.G., D.C.-M. and D.L.-d.-I.; methodology, J.G.G.; validation and formal analysis, J.G.G. and D.C.-M.; writing—original draft preparation, J.G.G. and D.C.-M.; writing—review and editing, D.C. and D.L.-d.-I.; supervision, D.L.-d.-I. and D.C.-M. All authors have read and agreed to the published version of the manuscript.

**Funding:** This research was funded by SentientThings grant number TIN2017-90042-R thanks to the Spanish Ministry of Science and Innovation.

**Acknowledgments:** All people that have helped with the survey, Grupo NEXT and the Ethical Deusto Commission. Co-authors Diego Casado-Mansilla and Diego López-de-Ipiña are grateful to project SentientThings granted by Spanish Ministry of Science and Innovation.

**Conflicts of Interest:** The authors declare no conflict of interest.

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
