# Peer review of "Analysis of Driver’s Reaction Behavior Using a Persuasion-Based IT Artefact"

_sustainability, doi:10.3390/su12176857_

Round 1
Reviewer 1 Report
The paper is appropriate and relevant to the Sustainability Journal themes, providing some analysis on driver’s reactions using an existing Persuasion-based IT artefact to optimize driver costs.
According to the authors, the research on Persuasion based IT artefacts for road safety purposes is already extestensive, but the research on how to apply those artefacts to optimize driver costs is less developed.
The focus is on an IT artefact called DAS, andhe methodology is based on a survey that had 301 responses, where DAS end-users self-rated possible persuasion principles/strategies.
Research objectives and methodologies are clear, and the paper is well enough structured.
I suggest to accept the paper after some revisions:
- I suggest to increase the readability of the graphs (figure 4 and 5) substituting the question number with a short statement that can help in recalling the objective of each question;
- I suggest to add in the conclusions a discussion on the possible weaknesses and threats of the proposed approach, maybe also highlighting possible ethical concerns in the use of persuasion based artefacts
- Finally, the paper needs an English check, since there are some grammar typos.
Author Response
Response to Reviewer 1 Comments
Point 1: The paper is appropriate and relevant to the Sustainability Journal themes, providing some analysis on driver’s reactions using an existing Persuasion-based IT artefact to optimize driver costs. According to the authors, the research on Persuasion based IT artefacts for road safety purposes is already extensive, but the research on how to apply those artefacts to optimize driver costs is less developed. The focus is on an IT artefact called DAS, and the methodology is based on a survey that had 301 responses, where DAS end-users self-rated possible persuasion principles/strategies.
Response 1: Thank you very much. In the new version of the manuscript, we have added information where we related the “saving expenditures” term with a sustainability vision, because having a less expensive car means to have a more sustainable car, i.e. less fuel consumption implies lower CO2 emissions.
Line Nº 43, 86
Point 2: I suggest to increase the readability of the graphs (figure 4 and 5) substituting the question number with a short statement that can help in recalling the objective of each question.
Response 2: Thank you for these comments. In the new version, this has been corrected and updated with a short statement for each question. We have added the following statement:
|
Nº |
Question |
Short Statement |
Screen Shot |
|
1 |
The APP is suitable for all times of the day |
Time suitable |
5 |
|
2 |
The APP offers me "tailor-made" services and can be customized by the user |
Tailor-made services |
4 |
|
3 |
The APP allows end user to monitor the behaviour and that allows end user to better reach the goals |
Reach goals |
3 |
|
3 |
The more I use the APP, the more proposals I receive where I like to receive praises for getting savings |
Receive praises |
1 |
|
5 |
I appreciate that the APP reminds me things I can do or things that I have to do in real time |
Remind things |
1 |
|
4 |
The way in which the proposals, as a sophisticated system, come to me, it conditions my response |
Response conditioned |
5, 1 |
|
7 |
The APP shows me a recognition and allows me to compare with other users |
Users comparison |
2 |
Line Nº 347, 359, 385, 397
Point 3: I suggest to add in the conclusions a discussion on the possible weaknesses and threats of the proposed approach, maybe also highlighting possible ethical concerns in the use of persuasion based artefacts.
Response 3: Thank you for the suggestion. We really think that it is a good point we missed and we have improved further the conclusion section, accordingly. Naturally, whoever creates the technology (designers, developers, engineers, etc.) has the power to incorporate elements that can condition to the end user. However, we consider persuasive technology as a means to nudge end-user without using coercion. It is true that this study has some limitations: we have little information about socio-demographic profiles of survey responders, even it is certainly a representative sample, and it does not cover the entire population that uses the artefact. We have modified the conclusion based on this concept and provided a limitations subsection. On the other hand, regarding the ethical considerations, the survey was launched once we had Deusto University’s Ethics Commission approval and everything was done supervised by the University rules and observing European GDPR.
Line Nº 312, 467
Point 4: Finally, the paper needs an English check, since there are some grammar typos.
Response 4: In the new version, this has been corrected by a native English speaking person.

Reviewer 2 Report
Authors of the paper present the results of a survey where the respondents were asked to rank different principles of persuasion to evaluate the most relevant for them to save time and money with their car. The subject of the paper is interesting and worth investigating. The methodology of the research is understandable, but some issues require further clarification.
What functionalities the authors have in mind. Is it, for example, indicating the shortest route and providing information on time or fuel savings? Is it providing other information (please indicate what kind of information?).
Shouldn't the detailed functionalities and type of information provided influence the results?
The article presents a very general approach. Please explain if there is no concern that the respondents understood the questionnaire correctly? Didn't the type of functionality provided by the application influence the respondents' assessment?
It would be reasonable to describe in more detail what information is provided to drivers and to comment on the impact of the type of information provided on the assessment made by respondents.
Author Response
Response to Reviewer 2 Comments
Point 1: Authors of the paper present the results of a survey where the respondents were asked to rank different principles of persuasion to evaluate the most relevant for them to save time and money with their car. The subject of the paper is interesting and worth investigating. The methodology of the research is understandable, but some issues require further clarification.
Response 1: Thank you for your comments. We have tried to address all your concerns improving the comprehensibility of the overall manuscript, as described in the answers to the following remarks received.
Point 2: What functionalities the authors have in mind. Is it, for example, indicating the shortest route and providing information on time or fuel savings? Is it providing other information (please indicate what kind of information?).
Response 2: Thank you for these comments. In the new version of the manuscript, your remarks have been included, updating the paper with detailed system information to improve the understandability of the proposed idea. The functionalities are focused on suggesting ad hoc services, when needed, with better pricing in comparison with the market. Furthermore, the system also provides other information related to the car and mobility. We divide the latter information in two categories:
- Car information: Location, speed, car failures, engine check status, car movements, fuel, trip statistics, maintenance schedule, insurance schedule.
- End user (Driver) services (depending on the real time notifications from the device to the server, and from the server to the APP): If the system collects an alert, depending on the user profile or the car needs, a message is going to be sent in real time to the user with recommendations for the car (stop for tank refuel, car maintenance or insurance renewal, for instance) and for the driver (In long trips, system recommends stops, parking, restaurants, gyms, hotels, and so on).
Line Nº 159
Point 3: Shouldn't the detailed functionalities and type of information provided influence the results?
Response 3: Thank you for your comment. The responders had been using the DAS (Device Artefact System) in their vehicles for 7 months before they were surveyed, so they are familiarized with the service provided. In a daily basis 55% of the users open the app. Weekly, 65% of the users use the app and monthly more than 80%. According to these figures, we can confirm that the full user sample should know what we are talking about when proposing to them the questionnaire. We have included these facts in the new version of the manuscript.
In any case, what we aim to analyse in the manuscript is which of the strategies help to develop a better system with the focus of creating a better solution which can guide better drivers reducing the costs and lifetime of their cars. The respondents answered, over all, about things that they have seen and have tried in the APP. For that reason, the main idea of the study was to understand how drivers rank the different persuasive strategies (some of them already implemented in the App), more than discovering new functionalities. In essence, we believe that the Hawthorne effect in which individuals modify an aspect of their behaviour in response to their awareness of being observed was removed during the first month over the use of the App. Therefore, our consideration is that responses received are free for such conditioning effect.
Line Nº 300, 332
Point 4: The article presents a very general approach. Please explain if there is no concern that the respondents understood the questionnaire correctly? Didn't the type of functionality provided by the application influence the respondents' assessment?
Response 4: Thank you for these comments and questions. We reflected following an inter-rater approach among the three authors of this paper on the kind of questions we should ask to the respondents, and we finally decided to include as simple as possible questions to maximize the quality of the answers received, i.e. the shorter and briefer the questionnaire the higher the chance that users will answer it and also dedicate enough time to complete it. The survey was thought to rank persuasion strategies based on persuasion principles. Those answers were elaborated thinking on the current DAS functionality and the associated mobile App. As we have commented in the previous point, the respondents are experts on the DAS system since they have been users of it for 7 months before this study. For that reason, we have eliminated unnecessary details in our survey and went straight to the point because of the prior knowledge of the users. We hope that this rationale can be self-explicative for the question related to the general approach.
As commented above, the type of functionality provided in DAS influences the respondents’ assessment. However, according to the results, the majority of final-users seem to prefer the persuasion strategies S3, S2 and S5. Please note that each participant in the survey responded the questions in a randomly sorted manner removing any kind of order bias. Therefore, we think their responses where free of influence, at least because of the questionnaire, and we can consider them valid.
To sum up, we consider that this paper addresses a review on what persuasion-based strategies are the most relevant to align the persuasion principles with future functionality additions. With usual limitations, we consider that it was an objective proposal lowering as much as possible the potential biases in this type of research. We have added some paragraphs in the paper to aid in better understanding of all of this.
Line Nº 332, 380
Point 5: It would be reasonable to describe in more detail what information is provided to drivers and to comment on the impact of the type of information provided on the assessment made by respondents.
Response 5: Thank you for your comment. In the new version of the manuscript, we have included the information provided by the DAS to the end user in terms of usability and messages. As we previously mentioned, the functionalities requested in the survey, currently exist. Please, note that the question 7, related to the comparison with peers – Social comparison principle, has been included to assess the interest of the end user. However, it is the sole functionality which is not yet activated in the App. We have included this limitation in the manuscript as it can induce some bias in the responses as usually we rate better something familiar that other components that we have never used before (Familiarity bias is the preference of the individuals to remain confined to what is familiar to them. They wish to remain within their comfort zone and refrain from taking the path never taken.).
Regarding the reviewer’s final comment. The main idea of this paper is to assess the ranking of the persuasion principles to help us to identify where we can improve the app with novel persuasion strategies. However, evaluating the impact of these strategies is considered as future work. We hope this will align with fewer resources used, money saved and overall reduction in CO2 emissions. We have included that on the conclusion section.
Line Nº 405, 448

Round 2
Reviewer 2 Report
The authors addressed all mentioned suggestions for improving the manuscript. In my opinion it has been improved. The only aspect left to be decided (but I consider this is the editor's opinion) is whether this paper fits into Sustainability scope.